# Development of Histologically Verified Thyroid Diseases in Women Operated for Breast Cancer: A Review of the Literature and a Case Series

**DOI:** 10.3390/jcm11113154

**Published:** 2022-06-01

**Authors:** Fausto Fama’, Alessandro Sindoni, Hui Sun, Hoon Yub Kim, Girolamo Geraci, Michele Rosario Colonna, Carmelo Mazzeo, Gabriela Brenta, Mariarosaria Galeano, Salvatore Benvenga, Gianlorenzo Dionigi

**Affiliations:** 1Department of Human Pathology in Adulthood and Childhood “G. Barresi”, University Hospital “G. Martino” of Messina, 98125 Messina, Italy; mrcolonna1@gmail.com (M.R.C.); dott.carmelomazzeo@gmail.com (C.M.); mrgaleano@libero.it (M.G.); 2Department of Public Health and Infectious Diseases, Sapienza University of Rome, 00185 Rome, Italy; alessandrosindoni@alice.it; 3New Hospital of Prato S. Stefano, Azienda USL Toscana Centro, 59100 Prato, Italy; 4Division of Thyroid Surgery, China-Japan Union Hospital of Jilin University, Jilin Provincial Key Laboratory of Surgical Translational Medicine, Changchun 130033, China; s_h@jlu.edu.cn; 5Department of Surgery, KUMC Thyroid Center, Korea University Hospital, Korea University College of Medicine, Seoul 02841, Korea; hoonyubkim@gmail.com; 6Department of Surgical, Oncological and Stomatological Sciences, University of Palermo, 90100 Palermo, Italy; girolamo.geraci@unipa.it; 7Division of Endocrinology, Cesar Milstein Hospital, Buenos Aires C1221 ABE, Argentina; gbrenta@gmail.com; 8Department of Clinical and Experimental Medicine, University Hospital “G. Martino” of Messina, 98125 Messina, Italy; s.benvenga@live.it; 9Division of Surgery, Istituto Auxologico Italiano IRCCS (Istituto di Ricovero e Cura a Carattere Scientifico), 20122 Milan, Italy; gianlorenzo.dionigi@unimi.it; 10Department of Pathophysiology and Transplantation, University of Milan, 20122 Milan, Italy

**Keywords:** breast cancer, thyroid disease, thyroid cancer, thyroid nodules

## Abstract

Background: The possible relationships between breast and thyroid diseases have been reported in the literature. The purpose of our study was to evaluate the occurrence of histologically verified thyroid pathologies in women who were diagnosed with breast cancer and, after mastectomy/quadrantectomy complemented by oncological treatment, were thyroidectomized based on their periodic thyroid evaluation. Patients and Methods: Our series consist of 31 women with a mean age of 62.9 ± 10.9 years (range, 45–81) treated for breast cancer (18 right-sided, 11 left-sided, and 2 bilateral), of whom 29 were thyroidectomized, since two women who developed Graves’ disease refused thyroidectomy. These 31 women belong to a cohort of 889 women who referred to the Breast Surgery Unit of our university hospital during the period January 2010 through December 2020. Results: The mean time interval between breast cancer and thyroid pathologies was 48.1 ± 23.4 months (range, 12–95). The final diagnosis at histopathology was infiltrating ductal breast carcinoma in 26 women (with 2/26 patients having bilateral carcinoma) and infiltrating lobular breast carcinoma in the other 5 women. Ten of the twenty-nine thyroidectomized women (34.5%) had a thyroid malignancy on histology: five papillary carcinomas, three papillary micro-carcinomas and two follicular carcinomas. Two of the five women with papillary carcinoma also had histological evidence of chronic lymphocytic thyroiditis/Hashimoto’s thyroiditis, which was also detected in another five women with benign thyroid diseases. Conclusions: We suggest that breast cancer survivors should be made aware of the possible increased risk of thyroid pathologies (including thyroid malignancy) so that they can undergo screening and follow-up.

## 1. Introduction

Breast cancer (BC) is the most common malignancy in females. Thyroid diseases, including thyroid cancer (TC), are more frequent in females than males [1,2].

Diagnosis and overtreatment of clinically insignificant malignancy is an enormous challenge for primary care providers, specialists, and patients enrolled in screening programs [3]. Second primary malignancies account for approximately 10–15% of new-onset cancers, and the number of individuals who have undergone cancer treatment at some point in their lives is growing by 2% annually [3]. The development of subsequent cancers can be attributed to a number of potential risk factors that include radiation therapy, chemotherapy, genetic predisposition, environmental factors, endocrinological changes, and impaired immune function [4].

The benefits of widespread ultrasound (US) screening are still controversial [4]. The systematic review and meta-analysis by Chen et al., demonstrated that hyperthyroidism, chronic lymphocytic thyroiditis/Hashimoto’s thyroiditis (HT) and TC are significantly associated with an increased risk of BC [4]. The sensitivity of mass screening for TC by neck US was also assessed in patients scheduled to undergo either a breast examination or a follow-up examination for BC [5]. Overall, thyroid nodules (TN) were detected in 25.2% of cases, with a 2.6% detection rate for TC. Not unexpectedly, in patients with TN detected by US in mass screening the size of TC was smaller than in patients with TN detected clinically [5]. Interestingly, a Chinese study [6] reported that nonmalignant TN were more common in women with BC than those without.

Neck US screening has led to a potential reduction in advanced disease and TC specific mortality, which is already generally low for TC. The incidence of TC, particularly of small and indolent TC, has nearly tripled since 1975, while mortality rates have remained largely unchanged [7]. Overtreatment is associated with well-documented morbidity that affects quality of life (e.g., postoperative recurrent laryngeal nerve injury and hypoparathyroidism) [2].

The purpose of our study was to evaluate the occurrence of histologically verified thyroid pathologies that developed after surgical and oncological treatments in women treated for BC and who were subsequently thyroidectomized. Moreover, we review the literature on patients who developed thyroid pathologies after treatment of BC.

## 2. Patients and Methods

This study involved 889 women who were referred to the Breast Surgery Unit of our university hospital during the period January 2010 through December 2020. All medical records of patients were retrospectively reviewed. The study was in accordance with the University Institutional Review Board and Helsinki Declaration. Written informed consent was obtained from all subjects.

All 889 women were preoperatively investigated by physical examination, breast US (7.5–14 MHz linear transducer), bilateral mammography and biopsy when required. Endocrine consultation ensured that a pre-existing thyroid disease could be detected. Surgical breast procedures were carried out on the basis of the images and reports of preoperative investigation. In all cases, an intraoperative and definitive histology assessment was done. Specimens were always orientated with threads and routinely stained. After surgical discharge, patients were examined in ambulatory until they were completely healed and they were also checked at 3, 6, and 12 months after surgery.

Once the 889 women were operated, 414 (46.6%) were diagnosed with histologically verified BC. As illustrated in Figure 1, of these 414, a total of 88 were excluded because having a thyroid disease without previous thyroidectomy (*n* = 58 [14.0% of 414]) or already thyroidectomized (*n* = 23 [5.5% of 414]). Moreover, 34 women [8.2% of 414] who underwent immunotherapy for HER2-positive BC were also excluded, considering the increased risk of developing autoimmune diseases, such as HT after this immunotherapy, could represent a potential bias. Since another five BC women (1.2% of 414) were lost to follow-up, the remaining cohort consisted of 294 women.

All 294 women received conventional adjuvant postoperative (chemotherapy, radiation therapy, and hormonal therapy in patients with hormone-sensitive BC) cancer treatments at the Oncology Unit of our university hospital. In particular, radiotherapy was traditionally performed with a mono-isocentric technique both on the breast (total dose of 50 Gy administered in 25 fractions, boosted by supplementary 10–14 Gy to the tumor bed) and on the axilla (46–50 Gy in 23–25 fractions on ipsilateral axillary apex, infraclavicular and supraclavicular lymph nodes). Furthermore, 86 of them (20.8%, 86/414) also received neoadjuvant chemotherapy.

Through the collaboration with the Endocrine Unit of our university hospital, all these women underwent a periodic thyroid evaluation at clinical level (history and physical examination), instrumental level (neck US with a 7.5–14 MHz linear transducer, with fine-needle aspiration cytology [FNAC] of TN and thyroid scintigraphy performed only if required) and biochemical level (measurement of serum thyrotropin (TSH), free thyroxine (FT4), free triiodothyronine [FT3], thyroperoxidase and thyroglobulin (TPOAb and TgAb) autoantibodies. Such endocrine evaluation occurred every 6 or 12 months thereafter, depending on whether a thyroid problem had been detected at previous check-ups. Cytological assessment was done according to Bethesda System for Reporting Thyroid Cytopathology which distinguishes the following categories: I = nondiagnostic or inadequate; II = benign, III = AUS or atypia of unknown significance/FLUS or follicular lesion of unknown significance, IV = suspicious for follicular neoplasm, V = suspicious for malignancy, and VI = malignant. Inclusion in the final group of patients to be analyzed required having completed at least the first 12 months of thyroid monitoring.

This retrospective analysis process allowed us to select a cohort of 31 patients operated for BC and who, also, developed a thyroid disease at least 1 year after breast surgery (10.5%, 31/294).

Of the 31 patients, 29 (93.5% of 31) underwent thyroid surgery because two women who developed Graves’ disease (GD) preferred medical therapy (methimazole) in lieu of thyroidectomy. Reasons for thyroidectomy were suspicious/frank malignancy at FNAC in 12 women and mechanical or esthetic issues in the remaining 17 women.

Continuous data are reported as mean ± standard deviation (range). Categorical data are reported as percentages, with statistical differences handled with the Fisher’s exact test upon setting the threshold of significance at *p* < 0.05.

Our literature review was conducted by searching the PubMed database using the following keywords: (breast cancer AND thyroid disease) OR (breast cancer AND thyroid nodule) OR (breast cancer AND thyroid cancer).

## 3. Results

Our study group comprised 31 women aged 62.9 ± 10.9 years (45–81). Of the 31 women, 29 had unilateral BC (18 in the right breast and 11 in the left breast) and the remaining two patients had bilateral BC. The distribution of lesions between the right and the left breast did not differ significantly (*p* = 0.398). Twenty-two patients underwent quadrantectomy (*n* = 21 unilateral, *n* = 1 bilateral) with axillary lymph-node dissection (ALND), and nine had modified Madden mastectomy (*n* = 8 unilateral, *n* = 1 bilateral). The final histological diagnoses of the 33 BC in the 31 patients were infiltrating ductal BC (ID-BC, *n* = 28) and infiltrating lobular BC (IL-BC, *n* = 5) (Table 1). Mean hospital stay was 3.8 ± 1.3 days (1–7) after surgery.

Two patients with GD preferred continuing to be treated with medical therapy (see above), so that 29 of the 31 patients were thyroidectomized. Thyroid FNAC was performed in 19/29 (65.5%) patients. The cytological assessment, in accordance with the Bethesda system, classified TN as Category II (7 cases), III (5 cases), IV (3 cases), V (3 cases) and VI (1 case). The intervention consisted of total thyroidectomy alone (*n* = 24) or associated with central neck lymph-node dissection (*n* = 5). Mean time interval between the histological diagnosis of BC and that of thyroid disease was 48.1 ± 23.4 months (12–95).

Final histological diagnoses of the thyroid lesions were colloid cyst multinodular goiter (MNG, *n* = 15), follicular adenoma (FA, *n* = 4), and TC (*n* = 10). Malignancy consisted of papillary thyroid carcinoma (PTC) having a maximum diameter greater than 10 mm (macro-PTC, *n* = 5), PTC having a maximum diameter up to 10 mm (micro-PTC, *n* = 3), and follicular thyroid carcinoma (FTC, *n* = 2) Colloid/cystic nodules associated with MNG occurred in the two patients with bilateral BC, in 7/18 patients with right-sided BC (38.9%) and in 6/11 patients with left-sided BC (54.5%) with no statistically significant difference between these two proportions (*p* = 0.745). The maximum diameter of the colloid/cystic nodules was 32.1 ± 8.2 mm (20–52), and they were localized in the right lobe (*n* = 11) and in the left lobe (*n* = 7). Histologically, HT was found in 7/29 women (24.1%), and it was associated with colloid cyst MNG (*n* = 3), macro-PTC (*n* = 2), and FA (*n* = 2) (Table 2).

The four FA had a maximum diameter ranging from 19 to 32 mm (24.75 ± 5.9), and they were localized in the right lobe (*n* = 2) or in the left lobe (*n* = 2). They occurred (right lobe, *n* = 2, and left lobe, *n* = 1) in 3/18 patients with right-sided BC (16.7%) and 1/11 patients (left lobe, *n* = 1) with left-sided BC (9.1%).

The two patients (2/31) with bilateral BC had colloid cyst MNG without other associated thyroid diseases. The two FTC were located in the isthmus (28 mm of maximum diameter) or in the left lobe (22 mm), and they occurred in patients with right-sided or left-sided BC, respectively. The five macro-PTC had a maximum diameter ranging from 12 to 25 mm (18.8 ± 5.4), which were always localized in the right lobe. Four of the five macro-PTC occurred in 4/18 patients with right-sided BC (22.2%), and the remaining one in 1/11 patients with left-sided BC (9.1%) (*p* = 0.410). The three micro-PTC had maximum diameter ranging from 4 to 7 mm (5.3 ± 1.5), and were localized in the right (*n* = 1) or in the left (*n* = 2) lobe. The three micro-PTC occurred in 1/18 patients (5.5%) with right-sided BC (micro-PTC in the right lobe) and in 2/11 patients with left-sided BC (18.2%, micro-PTC both in the left lobe) (Table 1).

Regarding the relationship between histological and cytological diagnosis, the five macro-PTC had been diagnosed as category II in two cases, category III in one case and category IV in two cases while both FTC had been diagnosed as category IV. The three micro-PTC were incidental. Relationships between the immunohistochemical status of BC and thyroid pathologies are highlighted in Table 3. Interestingly, 4/5 (80%) IL-BC were PR negative and 3/5 (60%) had more than 10% of BC cells positive at MIB-1 immunohistochemistry. Furthermore, in all 5 cases of IL-BC no HT was histologically detected; in 4/5 cases, the thyroid lesion was benign (colloid cyst MNG), while in the remaining one it was malignant (micro-PTC of 7 mm).

With the limitation given by the size of our series, some interesting data appear from close inspection of Table 3. One is that the most represent immunohistochemical group (BC ER+ PR+ MIB-1 ≤ 10%; *n* = 18) encompasses all thyroid diagnoses. Another is that colloid cyst MNG falls in all five immunohistochemical groups, while the next still benign lesion, FA, falls in 3/5 immunohistochemical groups. In contrast, thyroid malignancy is more restricted, particularly FTC, with both FTC cases falling in one immunohistochemical group.

## 4. Discussion

The risk of developing breast and thyroid pathologies depends on the characteristics of the subjects and the probability of identifying them from the diagnostic techniques used. In our previous study, we analyzed the possible relationships between non-cancerous breast nodules (BN) and TN, collecting evidence that can be taken into account in the clinical practice [7]. In a cohort of 127 women that underwent both thyroid and breast US scans, two-thirds of them (*n* = 84) had simultaneous cystic BN and the remaining one-third (*n* = 43) had solid BN [7]. Furthermore, cystic BN, but not solid BN, were more likely to be located on the same side of TN. Cystic BN were more often associated with multiple TN (64% of cases). In women with solid BN, the TN were larger and located in the middle part of the thyroid lobes, while in those with cystic BN, the thyroid lesions were smaller and located in the lower part of the thyroid lobes [7].

A Spanish study evaluated the association between various thyroid pathologies and the density of breast tissues on mammograms, the density depending directly on the proportion of stroma-epithelial tissue, and inversely on the proportion of adipose tissue [8]. The authors investigated 2883 women aged 47 to 71 years who participated in various screening programs, reporting a percentage density of breast tissue greater than 75% increases the risk of cancer by about six times, compared to percentages of less than 5% [8]. Among these patients, 13.9% reported having thyroid dysfunction or thyroid disease. In patients with nonautoimmune goiter, the density of breast tissue was reduced while in those with HT, it was increased [8].

BC and TC are common malignancies in women [9] and their coincidence was found to be associated with an increased long-term survival [10].

Literature shows an association of BC and TC diagnoses regardless of which cancer was first diagnosed. Patients with BC had a 10–30% increased risk of TC, while women with primary TC had a 5–20% increased risk of BC diagnosis compared with the general population [7,11]. In both cases, the increased risk of identifying a second primary malignancy was greatest shortly after the primary cancer diagnosis [3,4,7,8,11].

The risk of developing a second malignancy is greater in both BC and TC patients. This bidirectional relationship is reported in the medical literature although the underlying causes are unknown. Bolf et al. [12] hypothesized that chemotherapy and radiotherapy of the primary tumor, genetic variants, hormonal signaling, and lifestyle and environmental factors can contribute. The authors reviewed eight articles regarding the possible link between the incidence of breast and thyroid disease [12]. Among these, two articles analyzed only breast diseases developed after thyroid diseases, while another one evaluated the incidence of TC in patients with benign breast diseases. The remaining five articles, which were published between 2013 and 2017, analyzed the occurrence of thyroid disease in BC survivors. In detail, four retrospective papers evaluated the metachronous onset of TC and found that women with BC are at least 2 times more likely to develop future TC than the general population. The fifth and solely prospective study found a 2.5-fold greater incidence of TN in patients with a history of breast disease, including BC, compared to the general population [12].

In a previous review conducted by Nielsen et al., 19 studies published in the period 1984–2013, all related to the period 1935–2007 and with more than 1000 subjects included, evaluated second primary neoplasms as a whole following BC [11]. In all these large studies, the standardized incidence ratios of various second primary tumors were described and compared with relative risk and expected numbers. However, few details on the second tumors were provided, such as the finding that the lengthening of the median life, including women who has BC first, provided an increased risk of developing a second tumor, including TC, compared to the general population [11]. Focusing on the incidence of TC in BC survivors, five of these papers presented a number of second primary TC > 50 cases, providing inconclusive results regarding the influence of the time of BC diagnosis and the role of chemo- and radiation therapy received after breast surgery [11].

Several hypotheses may explain the apparent association between two distinct cancers, such as genetic or environmental risk factors, or treatment effects [3,11]. Moreover, surveillance of individuals who received cancer treatment influences subsequent cancer detection, a phenomenon known as surveillance bias. It occurs when patients receive more diagnostic tests or undergo a closer follow-up than others, resulting in more frequent diagnostic findings. Additionally, cancer survivors are more likely to attend cancer screenings compared with the general population, resulting also in the diagnosis of cancers with limited clinical significance. Such surveillance bias can contribute to the diagnosis of other tumors in patients affected by TC [11,13]. We have to notice that cancer survivors are not the only responsible for the increased access to additional screening, but also providers can play an important role in the management of these patients. Moreover, cancer screening services are more likely to be offered when oncologists and primary care physicians acts together [13].

A number of studies report rare cases of thyroid metastases from BC or, less frequently, breast metastases from TC. Thyroid metastases should be suspected in patients with a thyroid nodule and a previous history of non-thyroid malignancy; their prognosis is poor, but surgical management may be helpful [14]. Several authors suggest that pre-existing thyroid disease (such as MNG and TN) may provide a nidus for metastatic thyroid gland [15].

A recent Chinese study of 12,538 women undergoing simultaneous US breast and thyroid examination showed a bidirectional relationship between breast mass (BM) and TN, meaning that women with BM are at increased risk for TN and vice versa. Furthermore, thyroid hormone, in addition to being correlated with the onset of TN, also influences the development of BM [16].

Interestingly, Shi et al., highlighted that the incidence of TN in women operated for both benign and malignant breast diseases is higher than in the normal Chinese population [6]. Furthermore, these authors found that serum levels of thyroxine in initially diagnosed BC patients were significantly higher than those in benign breast diseases patients [6].

The overall prevalence of thyroid disorders, mainly nontoxic goiter and HT, is increased in patients with ductal BC (almost 28% and 14%, respectively), such prevalence being independent from both the estrogen receptor (ER) and progesterone receptor (PR) status of the BC [17].

Del Rio et al., instead, found inconclusive results on the co-occurrence of BC and TC [18].

A retrospective study considered a heterogeneous group of neoplastic diseases (i.e., 23 patients with gastrointestinal cancer, 11 pts with BC, and 7 with other-site primary cancer) [19]. TN greater than 1 cm in maximum diameter that were incidentally discovered in patients with another malignant tumor tested malignant on biopsy with a rate of 24%, which is above the predicted rate of 5% observed in traditionally discovered TN, thus justifying a more accurate evaluation of these patients [19].

Table 4 highlights recently reported clinical series of women with BC who were followed up to detect thyroid pathologies.

Hormonal factors may be responsible for the association between breast disease and thyroid disease: in fact, a greater expression of ER has been described in the breast tissues of women who have developed both types of cancer compared to those who have only had BC. It has been shown that ER is expressed on both normal and malignant thyroid tissue, the latter at quantitatively greater levels than the former [20]. In particular, estrogens are associated with a greater invasiveness and metastatic capacity of TC cell lines [21]. These data point out that ER play a role in the development of TC and in the increased rates of association between TC and BC. Zervoudis et al. [22] hypothesized a possible hormonal pathway of these two malignancies since the hormonal change in women who had many children or abortions could be a risk factor to develop both cancers. Genetic predisposition in the co-occurrence of BC and TC could play an important role [23]. Considering mutations, a ten-year risk of developing TC was higher in women who carried a CHEK2 mutation (1.5%) with respect to women with no mutation (0.9%) [24] (Table 4).

On the other hand, several studies have focused on the role of iodine in BC. Iodine deficiency may stimulate the gonadotrophin secretion resulting in a hyper-estrogenic status, with a high production of estrone and estradiol. This alteration in endocrine state may increase the risk of BC, especially in young woman [25]. In contrast, excess iodine intake also plays an unfavorable role by stimulating ER-α transcriptional activity in BC cells, thus favoring its growth [26]. Radioiodine treatment for differentiated TC increases the risk of secondary BC in female patients [27].

**Table 4 jcm-11-03154-t004:** Comparison of our series with clinical series that were published after 2017.

Author, Year, Reference	Type of the Study (Years of the Cohort Collection)	Number of pts Evaluated with Thyroid Diseases Associated	Modality of Thyroid Disease Association Verification	Thyroid Disease Associated and Most Important Key-Points of the Articles
Our series	Retrospective, single-center (2010–2020)	31	Histology	Authors analyzed 31 pts out of 294 operated for BC and who developed a histologically verified thyroid disease at least 1 year later breast surgery (10.5%). Thyroidectomy was performed in 29/31 BC pts. Malignant thyroid disease as second primary tumor was found in 34.5% of pts (5 macro-PTC, 3 micro-PTC and 2 FTC). Seven of these 10 thyroid malignancies were both ER and PR positive. The most frequent final diagnosis (51.7%) was colloid cystic MNG, while HT was found in 7 pts (24.1%).
Cieszynska et al., 2022 [24]	Prospective, multicenter (1996–2014)	53	Histology	Among 10,792 BC pts, 53 pts (0.49%) developed TC during a mean follow-up period of 14 years, that is 4 times greater than the expected number of 12 pts. TC histology was available for 50 pts, and it was PTC (*n* = 45 [90% of 50]), FTC (*n* = 3 [6%]) or medullary TC (*n* = 2 [4%]). The median time from BC diagnosis to TC diagnosis was 6.3 years. A total of 914 BC pts (8.3%) carried a CHEK2 mutation while 502 BC pts (4.6%) carried a BRCA1 mutation. Among the 914 women with a CHEK2 mutation, there were 10 TC observed, but only one was expected.
Kim et al., 2021 [28]	Retrospective, single center (1973–2009)	39	Histology	Out of a total of 6150 pts surgically treated for well-differentiated TC during the study period, the authors retrospectively investigated all cases in which there had been a co-diagnosis with BC, finding 99 up to the end of 2012. Of these 99 pts with co-diagnosis, only in 75 cases it was possible to examine the formalin-fixed paraffin blocks related to BC. The histological features of the differentiated TC were indicated in 71/75 pts (65 PTC, 5 FTC, and 1 case with both histotype). The authors report that in 39/75 pts, BC occurred before or simultaneously to TC (BC/TC group) and in the remaining 36, it was diagnosed after TC (TC/BC group) but the time interval between the 2 malignancies is not specified. A possible role of ER and TR in the link between two neoplastic diseases was hypothesized, considering that their increased expression is associated with the occurrence of TC.
Del Rio et al., 2020 [18]	Retrospective, single center (2010–2016)	43	US Cytology	The aim of the study was to value the incidence of BC in pts with a personal history of differentiated TC and conversely, the incidence of differentiated TC in pts with previous BC within 5 years from the diagnosis of the first tumor. All pts with a previous BC underwent to neck US before the operation (2010) and 6 years after it (2016) while pts with a previous differentiated TC were evaluated with mammography screening. In this retrospective evaluation, the authors state that they have found inconclusive results on the co-occurrence of BC and TC. Only 43 BC were further considered (28 had ID-BC or IL-BC, 13 had in situ-D-BC or in situ-L-BC and the remaining had 2 other histotype); among these pts, neck US detected TN in more pts in 2016 than in 2010 (21 vs. 13 or 49% vs. 30%) as well as diagnoses of thyroiditis which were made predominantly in pts in 2016 (10 vs. 6 or 23% vs. 14%). Six FNAC were made in 2016 (all results were category II), but the authors do not provide clear diagnoses of thyroid disease.

**Abbreviations**: Patients (**pts**); Infiltrating Ductal Breast Cancer (**ID-BC**); Estrogen receptor (**ER**); Thyroid hormone receptor (**TR**); Thyroid nodule (**TN**); Multinodular goiter (**MNG**); Thyroid cancer (**TC**); Hashimoto thyroiditis (**HT**); Papillary Thyroid Carcinoma (**PTC**); Follicular Thyroid Carcinoma (**FTC**); Fine-needle aspiration cytology (**FNAC**); Ultrasound (**US**).

Moreover, the possible interaction between thyroid gland and BC may be explained by the presence of thyroid hormone receptors (TR) in BC. Increased TR-α and ER-α expression in BC patients may be associated with a greater risk of developing a TC [28]. It has been hypothesized that TR overexpression in BC patients, facilitates its development, both independently and in combination with estrogen [29]. Thyroid hormones modulate the cellular and cytokine content of the BC microenvironment [30]. Additionally, FT3/FT4 ratio predicts the presence of carcinoma in situ component, multifocal/multicentric tumors and lymphovascular invasion in pathological specimens: an increased serum FT3 level is associated with the presence of carcinoma in situ component, multifocal/multicentric tumors, increased proliferative activity and poor prognosis [31].

Autoimmune thyroid diseases are characterized by increased levels of TPOAb and TgAb, and an association between HT and BC has been reported previously [17,26,32,33]. In particular, the prevalence of these antibodies in BC patients is significantly higher than in healthy controls [34].

In our series, 10 of 29 thyroidectomized women (34.5%) and with a history of previous BC surgery had a histologically proven malignancy: 5 macro-PTC, 3 micro-PTC and 2 FTC. The concordance between cytological and histological examination was more reliable for FTC (Bethesda category IV in both cases) than macro-PTC (Bethesda category IV in 2/5 cases). Moreover, in agreement with other recent series [24,28], the most frequent histotype among patients with neoplastic thyroid disease in our study was papillary (80%, 8/10).

Regarding the association between the immunohistochemical status of BC and thyroid pathologies we underline that in most cases (3 macro-PTC, 2 micro-PTC, and 2 FTC), BC were ER/PR-positive, only 1 micro-PTC was ER-positive and PR-negative, while in 2 macro-PTC they were ER/PR-negative. Overall, the most frequent final diagnosis was colloid cystic MNG (51.7%, 15/29) and most of the thyroid pathologies were associated with a hormonal receptor positive BC (62.1%, 18/29), supporting the hypothesis of their influence on the development of the second TC.

Finally, in the context of autoimmune thyroid disease, the frequency of which in BC women was reported by previous studies to be increased [2,17], we found that histologically HT was present in three cases with colloid cyst MNG, in two with macro-PTC, and in two with FA. Furthermore, two women, who refused to be thyroidectomized, had GD (6.5%, 2/31), the other classical autoimmune thyroid disease.

The main limitation of our study is that presented data do not enable strong statistical analysis but are of descriptive nature, due to the small size of the series. However, our patients were recruited in a single center, following a rigorous and well-defined methodology to detect the thyroid pathology (in all cases histologically verified) subsequent to the mammary one. Underlying causes for the development of a second malignancy and the potential link between the breast and thyroid cancers would benefit from further discussions.

## 5. Conclusions

Although our initial small sample may represent a limitation of our study, BC survivors should be made aware of the possible increased risk of thyroid disease, including even TC. Therefore, we suggest that they be included in screening and follow-up programs according to standardized guidelines that include at least an US scan and a biochemical assessment of the thyroid function per year, even in the absence of symptoms. Several factors could explain the association between the two diseases, including control bias, but also biological and environmental factors. It is possible that there are also genetic factors that can contribute to this event, and it is important to fully evaluate this possibility.

## Figures and Tables

**Figure 1 jcm-11-03154-f001:**
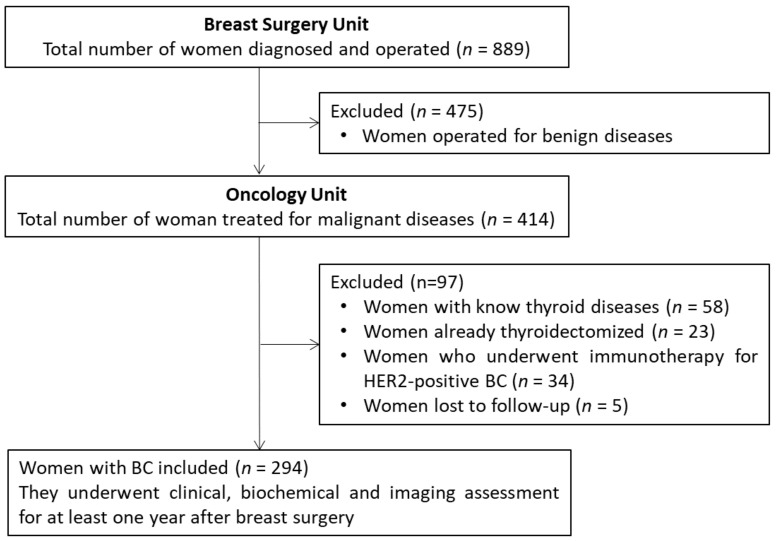
Flow chart of patient selection.

**Table 1 jcm-11-03154-t001:** Characteristics of BC of the women included in the study cohort.

Lateralityof BC (*n* = 31 pts)	BC Localizations (*n* = 33 Tumors)					Surgical Procedures (*n* = 33)		Immunohistochemical Features (*n* = 33 Tumors)			
	SEQ	SIQ	IEQ	IIQ	RA	Q + ALND (n = 23)	MM (+ALND) (n = 10)	ER+ >10%	PR+ >10%	MIB-1 ≤10%	MIB-1 >10%
**UR ID-BC (16 pts)** **UR IL-BC (2 pts)**	11	2	3	1	1	13	5	14/18	12/18	11/18	7/18
**UL ID-BC (8 pts)** **UL IL-BC (3 pts)**	6	2	0	1	2	8	3	9/11	8/11	7/11	4/11
**B ID-BC (2 pts)**	R1 R2 L2	L1	0	0	0	R2 L2	R1 L1	R1 L1 R2 L2	R1 L1 R2 L2	R2 L2	R1 L1

**Abbreviations**: Patients (**pts**); Breast cancer (**BC**); Unilateral (**U**); Bilateral (**B**); Right (**R**); Left (**L**); Infiltrating Ductal Breast Cancer (**ID-BC**); Infiltrating lobular Breast Cancer (**IL-BC**); Superior External Quadrant (**SEQ**); Superior Internal Quadrant (**SIQ**); Inferior External Quadrant (**IEQ**); Inferior Internal Quadrant (**IIQ**); Retroareolar (**RA**); Quadrantectomy (**Q**); Axillary Lymph Node Dissection (**ALND**); Madden Mastectomy (**MM**); Estrogen Receptor (**ER**); Progesterone Receptor (**PR**).

**Table 2 jcm-11-03154-t002:** Thyroid histological findings in the cohort of patients previously operated for BC who underwent thyroid surgery.

Thyroid Surgical Procedures (n = 29)	Colloid Cyst MNG (*n* = 15)	MNG with Macro-PTC (*n* = 5)	MNG with Micro-PTC (*n* = 3)	MNG with FTC (*n* = 2)	MNG with FA (*n* = 4)
**TT in pts with R-BC**	7 (3 §)	4 (1 §)	1	1	3 (1 §)
**TT in pts with L-BC**	6	1 (1 §)	2	1	1 (1 §)
**TT in pts with B-BC**	2	0	0	0	0

**Abbreviations**: Patients (**pts**); Total thyroidectomy (**TT**); Multinodular goiter (**MNG**); Papillary Thyroid Carcinoma (**PTC**); Follicular Thyroid Carcinoma (**FTC**); Follicular Adenoma (**FA**); Right-sided breast cancer (**R-BC**); Left-sided breast cancer (**L-BC**); Bilateral breast cancer (**B-BC**). **§** cases with histological Hashimoto thyroiditis.

**Table 3 jcm-11-03154-t003:** Relationships between immunohistochemical status of patients operated for BC and associated thyroid pathologies.

Immunohistochemical Status of BC (*n* = 33 Tumors)	Colloid Cyst MNG (*n* = 15)	MNG with Macro-PTC (*n* = 5)	MNG with Micro-PTC (*n* = 3)	MNG with FTC (*n* = 2)	MNG with FA (*n* = 4)
**ER+ PR+ MIB-1 ≤ 10%**	4 R ID-BC (2 §)	2 R ID-BC (1 §) 1 L ID-BC	1 R ID-BC 1 L ID-BC	1 R ID-BC 1 L ID-BC	1 R ID-BC (§) 1 L ID-BC (§)
	3 L ID-BC				
	1 L IL-BC				
	1 B ID-BC				
**ER+ PR+ MIB-1 > 10%**	1 B ID-BC				1 R ID-BC
**ER+ PR- MIB-1 ≤ 10%**	1 R IL-BC				
**ER+ PR- MIB-1 > 10%**	1 R IL-BC		1 L IL-BC		1 R ID-BC
**ER- PR- MIB-1 > 10%**	1 R ID-BC (§)	2 R ID-BC (1 §)			
	1 L IL-BC				
	1 L ID BC				

**Abbreviations**: Breast cancer (**BC**); Bilateral (**B**); Right (**R**); Left (**L**); Infiltrating Ductal Breast Cancer (**ID-BC**); Infiltrating lobular Breast Cancer (**IL-BC**); Estrogen Receptor (**ER**); Progesterone Receptor (**PR**); Multinodular goiter (**MNG**); Papillary Thyroid Carcinoma (**PTC**); Follicular Thyroid Carcinoma (**FTC**); Follicular Adenoma (**FA**). **§** cases with histological Hashimoto thyroiditis.

## Data Availability

All data were collected in a clinical medical record database. The database is maintained with quality assurance by an informatics specialist.

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
