# Peer review of "Development of Histologically Verified Thyroid Diseases in Women Operated for Breast Cancer: A Review of the Literature and a Case Series"

_jcm, 2022, doi:10.3390/jcm11113154_

Round 1

Reviewer 1 Report

The authors present an interesting case series regarding the occurance of thyroid diseases in women with breast cancer. They present a clearly structured and though-out manuscript.

However, the presented data does not enable statistical analysis but is of descriptive nature. Additionally, this manuscript does not introduce new and significant information.

Author Response

We thank the Reviewers for their comments and constructive criticism.

We have performed all suggested modifications in a point-by-point manner; changes are typed in red text. As a result, we believe that the quality of our paper has been significantly improved.

Reviewer #1:

The authors present an interesting case series regarding the occurrence of thyroid diseases in women with breast cancer. They present a clearly structured and though-out manuscript. However, the presented data does not enable statistical analysis but is of descriptive nature. Additionally, this manuscript does not introduce new and significant information.

We thank Reviewer # 1 for his/her comments. It is true that it is a small series of patients recruited in a single center but following a rigorous and well-defined methodology that culminated in the histologically-proven diagnosis of the thyroid disease. We respectfully disagree with the reviewer concerning his/her comment that “this manuscript does not introduce new and significant information”. We are unaware of other studies that have evaluated the association between the topographical localizations of the breast cancer and the thyroid nodule. Our retrospective study highlights the importance of shared follow-up protocols between surgery and endocrinology units.

Reviewer 2 Report

This paper presents the relationship of breast cancer patients and their incidence of developing thyroid cancers. The rates of breast nodules and thyroid nodules and their significance to the detection of potential breast and thyroid malignancies were well summarized. The significance of cystic nodules compared to solid nodules were well discussed. Comparison of study data to published series were clearly presented.  Although the small study population was small, this paper adds evidence to consider screening and follow-up protocols for thyroid cancer in patients with breast cancer.

The presented figure and tables are clear. The manuscript is well-structured and easy to follow. Cited references are appropriated. The content is scientifically sound with well thought out methodology. Conclusions are consistent and supported by the presented evidence. Ethics statements for the study are not explicit. 

Limitations of the study include: the underlying causes for the development of a second malignancy and the potential link between the breast & thyroid cancers would benefit from further discussions. Elaboration of the relationship of breast masses and thyroid nodules would add value. Additional hypothesis for the roles of estrogen receptors and thyroid hormone receptors would add strength to the paper. The role of iodine associated with breast cancer development warranted further development. Expansion on the exclusion of patients undergoing immunotherapy for HER2-positive breast cancer would be of benefit.

Overall, the paper adds meaningful data to the accumulating evidence for considerations of thyroid cancer screening for breast cancer patients.

Author Response

We thank the Reviewers for their comments and constructive criticism.

We have performed all suggested modifications in a point-by-point manner; changes are typed in red text. As a result, we believe that the quality of our paper has been significantly improved.

Reviewer #2:

This paper presents the relationship of breast cancer patients and their incidence of developing thyroid cancers. The rates of breast nodules and thyroid nodules and their significance to the detection of potential breast and thyroid malignancies were well summarized. The significance of cystic nodules compared to solid nodules were well discussed. Comparison of study data to published series were clearly presented. Although the small study population was small, this paper adds evidence to consider screening and follow-up protocols for thyroid cancer in patients with breast cancer. The presented figure and tables are clear. The manuscript is well-structured and easy to follow. Cited references are appropriated. The content is scientifically sound with well thought out methodology. Conclusions are consistent and supported by the presented evidence.

Overall, the paper adds meaningful data to the accumulating evidence for considerations of thyroid cancer screening for breast cancer patients.

We thank Reviewer # 2 for having appreciated our work.

Ethics statements for the study are not explicit.

Response: It is a retrospective study approved by the local ethics committee. Ethics statements are now detailed in the text (first paragraph of the Patients and Methods, lines 86-88).

Limitations of the study include: the underlying causes for the development of a second malignancy and the potential link between the breast & thyroid cancers would benefit from further discussions.

Response: The causes (genetic, environmental, hormonal, or immunological predisposing factors) of the development of second thyroid malignancies are not yet explained. As suggested by the Reviewer, we have added (last paragraph of the discussion, lines 378-380) the following sentence to the study limits paragraph: underlying causes for the development of a second malignancy and the potential link between the breast and thyroid cancers would benefit from further discussions.

Elaboration of the relationship of breast masses and thyroid nodules would add value.

Response: As part of our work, we have focused attention on the location and size of thyroid diseases following breast cancer. Therefore, the addition of the breast tumor sizes is out from the aims of our paper.

Additional hypothesis for the roles of estrogen receptors and thyroid hormone receptors would add strength to the paper.

Response: As stated in the text, the most accredited hypothesis that can still explain the association between thyroid and mammary neoplasms, both in terms of increased risk and tumor development, lies in the overexpression of thyroid hormone receptors in patients with breast cancer as well as the modulatory effect on the cytokines induced by the thyroid hormones themselves both at the intracellular level and on the mammary neoplastic microenvironment.

The role of iodine associated with breast cancer development warranted further development.

Response: The relationship between iodine and breast cancer has been covered in the discussion section. Recent studies retrieved in the literature show that the hyper-estrogenic state, i.e., the increased production of estrone and estradiol, can increase the breast cancer risk, particularly in young women, by increasing the receptor share and stimulating tumor growth itself.

Expansion on the exclusion of patients undergoing immunotherapy for HER2-positive breast cancer would be of benefit.

Response: Immunotherapy for the treatment of tumors, including HER2-positive mammary ones, by stimulating the lymphocytic immune response could favor the development of autoimmune diseases, such as thyroiditis, which in turn could increase the risk of secondary neoplasms; therefore, it was preferred to eliminate this potential bias.

A sentence concerning this comment has been added in the text (third paragraph of the Patients and Methods, lines 101-103).

Reviewer 3 Report

  1. Incidence of tyroid disease in breast cancer must be compered with incidence in general population.
  2. Date regarding the oncologic treatment  such as radiotherapy and chemotherapy must be detailed. Radiotherapy may influence too the incidence of tumor nodules in the tyroid,  so these information are important. 
  3. The sample size is small 

Author Response

We thank the Reviewers for their comments and constructive criticism.

We have performed all suggested modifications in a point-by-point manner; changes are typed in red text. As a result, we believe that the quality of our paper has been significantly improved.

Reviewer #3:

Incidence of thyroid disease in breast cancer must be compared with incidence in general population. The sample size is small.

We thank Reviewer # 2 for having appreciated our work. It is true that it is a small series of patients recruited in a single center but following a rigorous and well-defined methodology that culminated in the histologically-proven diagnosis of the thyroid disease. Our retrospective study highlights the importance of shared follow-up protocols between surgery and endocrinology units.

Date regarding the oncologic treatment such as radiotherapy and chemotherapy must be detailed. Radiotherapy may influence to the incidence of tumor nodules in the thyroid, so these informations are important.

Response: It is known that ionizing radiation can harm the thyroid. Radiation treatment was traditionally performed with a mono-isocentric technique, with a delivered total dose of 50 Gy in 25 fractions, followed by an additional boost of 10-14 Gy to the breast tumor bed; moreover, 46-50 Gy in 23-25 fractions can be delivered to ipsilateral axillary apex, infraclavicular and supraclavicular nodes. An informative sentence about radiotherapy has been added in the text (fourth paragraph of the Patients and Methods, lines 109-113).
